# Potential tradeoffs between effects of arbuscular mycorrhizal fungi inoculation, soil organic matter content and fertilizer application in raspberry production

**Ke Chen** \* , **Jeroen Scheper, Thijs P. M. Fijen, David Kleijn**

Plant Ecology and Nature Conservation Group, Wageningen University, Wageningen, The Netherlands

\* chenkejy@gmail.com

**Data Availability Statement:** All relevant data are within the paper and its Supporting Information files.

## Abstract

Ecological intensification has been proposed as an alternative paradigm for intensive agriculture to boost yield sustainably through utilizing ecosystem services. A prerequisite to achieving this is to understand the relations between multiple ecosystem services and production, while taking growth conditions such as nutrient availability into consideration. Here, we conducted a pot-field experiment to study the interactive effects of soil organic matter (SOM) content and arbuscular mycorrhizal fungi (AMF) inoculation on the production of raspberry (*Rubus idaeus* L.) under four levels of fertilizer application. Raspberry flower number, fruit number and yield only significantly increased with fertilizer inputs but were not impacted by SOM content or AMF inoculation. Fruit set and single berry weight were influenced by both SOM content and AMF inoculation, in complex three-way interactions with fertilizer application. Fruit set of AMF inoculated plants increased with fertilizer inputs in low SOM soils, but decreased with fertilizer inputs under high SOM soils, with the highest fruit set occurring at no fertilizer inputs. In low SOM soils, the relation between single berry weight and fertilizer application was more pronounced in inoculated plants than in non-inoculated plants, while in high SOM soils the relative benefits of AMF inoculation on single berry weight decreased with increasing fertilizer inputs. We attribute the lack of effects of AMF inoculation and SOM content on flower number, fruit number and yield mainly to potential tradeoffs between the experimental variables that all influence resource uptake by plant root systems. Our results suggest that potentially beneficial effects of AMF and SOM can be offset by each other, probably driven by the dynamic relations between AMF and the host plants. The findings reveal fundamental implications for managing AMF inoculation and SOM management simultaneously in real-world agricultural systems.

## Introduction

Conventional agricultural intensification cannot meet the twofold challenge facing agriculture: increasing yield to feed the growing world population while minimizing negative externalities

**Funding:** Ke Chen was financially supported through the China Scholarship Council (File No. 201706990023). The funder had no role in study design, data collection and analysis, decision to publish, or preparation of the manuscript.

**Competing interests:** The authors have declared that no competing interests exist.

on the environment [1,2]. It is increasingly difficult to further promote productivity through mainstream intensive farming practices [3,4], because the production is increasingly limited by critical natural ecosystem services, such as insect pollination [5,6] and soil formation [7]. Additionally, these intensive farming practices have caused severe environmental problems, such as soil and water pollution [8,9] and biodiversity loss [10,11], which are threatening human-wellbeing [3]. Ecological intensification has been proposed as a promising alternative for conventional intensive agriculture. It is based on managing multiple ecosystem services to complement and/or replace artificial inputs to maintain or enhance productivity while reducing negative environmental impacts [12,13]. Ecological intensification has been advocated as an environmentally friendly way towards food security [14,15] and an increasing number of studies provide proof of concept for this paradigm [16–18]. There are still knowledge gaps between theory and practice, however, which limit the adoption of ecological intensification by the agricultural sector [13]. For example, when multiple ecosystem services are managed in conjunction, their effects on production could interact synergistically, negatively or not at all [19,20]. Understanding whether and how different ecosystem services interact in shaping crop production is of importance to maximize the benefits of ecological intensification and promote its adoption [13].

Soil organic matter (SOM) and arbuscular mycorrhizal fungi (AMF) are two natural factors that provide or influence vital ecosystem services in cropping systems [12,17,21]. SOM is often used as a proxy for soil services, as it is able to mediate the flow of soil ecosystem services [12,22], and it strongly affects almost all soil properties [23]. Examples include soil structural stability and water-holding capacity (physical properties), cation exchange capacity and pH regulation (chemical properties), and nutrient supply for microbial communities (biological properties) [23]. SOM content, therefore, often relates positively to crop production [17,24].

AMF are widespread soil microorganisms from the phylum *Glomeromycota*, and they can form symbiotic associations with the majority of the cultivated crops [25,26]. AMF develop an extensive hyphal network through proliferating their hyphae inside plant roots (intracellular hyphae) as well as within the soil (extraradical hyphae), thus acting as a bridge between plant and soil [27–29]. AMF mainly help their host plants exploit poorly mobile ions (notably inorganic phosphate) that are beyond the root zone, in exchange for photosynthetic products from the host for metabolic needs [29]. Besides assisting with resource uptake, AMF colonization can also benefit the hosts by enhancing their tolerance to abiotic and biotic stresses, such as drought, salinity, diseases and pathogens [30,31]. Indirectly, AMF can benefit the hosts via improving soil structure and soil aggregation [32]. Inoculation of AMF has been found to promote crop yield [21,33,34], especially where the indigenous AMF communities have been degraded by agricultural practices [35,36].

A wealth of studies have shown that AMF and SOM can influence each other [37,38]. AMF are able to positively influence SOM content directly, through producing glomalin-related soil proteins [39,40], which are significant components of SOM [41,42]. Additionally, AMF has been found to affect the decomposition of SOM negatively [38] or positively [43]. On the other hand, various organic compounds released from the decomposition of SOM have been shown to influence AMF growth and activity, either positively [44] or negatively [45]. However, as far as we know, so far no studies ever clearly tested whether and how their effects on crop production interact. Furthermore, agricultural practices, in particular artificial fertilizer application, can influence the effects of both AMF [46] and SOM [24] as it also influences nutrient availability of crop plants. It is therefore essential to take fertilizer inputs into consideration when test the interacting effects of AMF and SOM on crop production. Here, we examined (1) the combined effects of AMF inoculation and SOM content on the production of raspberry (*Rubus idaeus* L.) and (2) how they are affected by fertilizer application.

## Materials and methods

### (a) Study system

Raspberry was used as the study crop, which is an important perennial fruit crop, with growing consumer interest due to its health benefits and flavours [47,48]. We selected the commercial cultivar *'Tulameen'*, as it is among the most popular raspberry cultivars in a range of climatic conditions [49] and is locally available. The study was conducted in an experimental field of Wageningen University & Research in the Netherlands, from August 2019 to September 2020.

### (b) Experimental setup

We adopted a randomized complete block design to account for potential confounding gradients in the experimental field, and we combined all of the following three crossed factors: (i) low SOM content vs high SOM content, (ii) AMF inoculated vs non-inoculated and (iii) four levels of fertilizer application (i.e. 16 plants per block). The SOM treatments were obtained by mixing different proportions of two types of sandy soils which had different SOM content (0.3% vs 4.6%) resulting in either 1.95% SOM content soils ('low SOM' treatment; available N: 14.0 mg/kg, available P: 0.6 mg/kg, available K: 19.4 mg/kg, pH: 6.6) or 3.96% SOM content soils ('high SOM' treatment; available N: 43.1 mg/kg, available P: 0.6 mg/kg, available K: 26.6 mg/kg, pH: 5.9). As for AMF treatments, we used *Rhizophagus intraradices* inoculum (MYKOS® Xtreme Gardening, Canada: 300 propagules/gram). Half of the original inoculum was autoclaved at 121˚C for two hours as sterilized inoculum for non-inoculated treatments [50]. The four levels of fertilizer treatments represented the equivalent 0, 33, 66 and 99 kg ha$^{-1}$ of N per year, ranging from no to optimum N inputs [51]. The fertilizer used was a compound fertilizer (Fertilizers®Cropsolutions, The Netherlands), containing 10.80% N, 13.44% K and 5.89% P.

We purchased 160 raspberry cuttings from a local supplier, with an average height of ca. 60 cm. To avoid the influence from the original peaty substrate, we carefully washed away the soil adhering to the roots in early August 2019. We added the recommended dose of AMF inoculum (25 grams) or an equal volume of sterilized inoculum evenly to the washed roots of the plants. The plants were then transplanted to a 10-litre plastic pot (upper diameter 28 cm, holes in the bottom for drainage but covered with cloth to minimize root growth out of the pot) and filled with low or high SOM soils according to the experimental design. However, higher than expected mortality occurred, possibly due to the cuttings being damaged during the roots washing process combined with the late summer heat. Only 56 plants survived out of the 160 plants, and 48 of them were of good health and thus were selected for further experimentation in three blocks. To carry out the experiment with sufficient replication, we additionally purchased another 160 raspberry cuttings in early October 2019. Because 60 cm cuttings were no longer available, we used plants with an average height of ca. 25 cm. Strictly following the earlier described protocol and using the same materials, the new batch of cuttings were washed, inoculated and transplanted into the low or high SOM soils. In this round, 110 out of the 160 new cuttings survived. These 110 plants were arranged into seven blocks. In total, the experiment therefore started out with 158 potted raspberry plants in 10 blocks. Plants with different treatments were placed randomly within each block. Plants were spaced with one meter within and between rows. Pots were dug into the soil to protect the roots from extremely high or low temperatures. The fertilizer treatments were applied by splitting the annual dose (0, 33, 66 and 99 kg ha$^{-1}$ of N) into three applications: the first one in the autumn, the second one at bud break in early spring of the following year and the last one at early flowering. All plants received equal and ample irrigation (depending on the weather conditions), and weeds in the

pots were regularly removed. Prior to berry ripening, all plants were bagged with mesh bags to avoid predation by animals. We harvested and weighed the ripe berries when they had just turned bright red. We summed up the berry weight from the same plant to get the total yield and fruit number. Additionally, we carefully counted the wilted or aborted flowers that failed to develop into fruits, which in combination with the fruit number allowed us to estimate the flower number. Because we could only have taken root samples at the end of the experiment, and earlier analyses showed AMF colonization rate of raspberry plants from different treatments did not differ after almost a year's growth [52], we did not measure AMF root colonization rates.

## (c) Data analysis

Until harvest, 41 plants from the first batch survived and developed fruits; all 110 plants from the second batch survived, but only 25 of them developed fruits. Since we mainly focus on the effects of treatments on production, only the plants that produced fruits were involved in the data analysis (sample size n = 66, S1 Table). We ran separate linear mixed-effects models using the function lme() of the nlme package in R [53] to study the interacting effects of SOM, AMF and fertilizer on flower number, fruit number, single berry weight (g/fruit) and total yield (g/plant), and included "block" as a random factor. We included the origins of the plants as a covariable in all models, to account for differences between plants from the first and the second batch. Because the fruit set followed a binomial distribution, we used the function glmmTMB() to run the same models assuming a binomial distribution [54]. Single berry weight was averaged per plant to avoid pseudoreplication, and response variables were transformed if necessary to meet the normality and homoscedasticity assumptions of the models.

Full models were simplified by removing non-significant predictors (backward elimination) using likelihood ratio tests with removal thresholds of p > 0.05, until the resulting minimum adequate model consisted only of variables that contributed significantly to the outcome [55,56].

## Results

The number of flowers per plant was only influenced by fertilizer inputs (Table 1). Plants receiving 99 kg N·ha$^{-1}$ produced 32% more flowers than plants without any fertilizer inputs (Fig 1A). Similarly, fruit number and total yield per plant were only affected by fertilizer inputs (Table 1). The fruit number of the plants grown with the highest fertilizer inputs was 69% higher than that of plants receiving no fertilizer (Fig 1B). Increasing fertilizer inputs from 0 to

**Table 1. Effects of arbuscular mycorrhizal fungi (AMF; inoculated vs non-inoculated), soil organic matter (high vs low SOM content) and fertilizer application rates (0, 33, 66, 99 kg N·ha$^{-1}$·year$^{-1}$) on raspberry fruit production variables (n = 66).** Bold values represent significant effects (P<0.05).

| | Flower number (sqrt transformed) | | Fruit set | | Fruit number (ln transformed) | | Single berry weight | | Yield (ln transformed) | |
|---|---|---|---|---|---|---|---|---|---|---|
| | $\chi^2_{(1)}$ | P | $\chi^2_{(1)}$ | P | $\chi^2_{(1)}$ | P | $\chi^2_{(1)}$ | P | $\chi^2_{(1)}$ | P |
| AMF | 0.185 | 0.667 | 1.614 | 0.204 | 0.232 | 0.630 | 2.070 | 0.150 | 0.005 | 0.943 |
| SOM | 0.601 | 0.438 | 22.136 | **0.000** | 0.936 | 0.333 | 1.304 | 0.254 | 0.137 | 0.711 |
| Fertilizer | 5.107 | **0.024** | 23.883 | **0.000** | 6.433 | **0.011** | 10.593 | **0.001** | 14.914 | **0.000** |
| Origin | 29.620 | **0.000** | 17.136 | **0.000** | 27.739 | **0.000** | 13.807 | **0.000** | 28.936 | **0.000** |
| AMF:fertilizer | 0.014 | 0.907 | 13.303 | **0.000** | 0.671 | 0.413 | 0.670 | 0.413 | 0.241 | 0.624 |
| AMF:SOM | 0.033 | 0.855 | 1.054 | 0.305 | 0.292 | 0.589 | 3.356 | 0.067 | 0.246 | 0.620 |
| SOM:fertilizer | 1.132 | 0.287 | 0.715 | 0.398 | 0.008 | 0.928 | 0.073 | 0.787 | 0.019 | 0.891 |
| AMF:SOM:fertilizer | 0.768 | 0.381 | 16.053 | **0.000** | 0.047 | 0.829 | 4.722 | **0.030** | 1.438 | 0.230 |

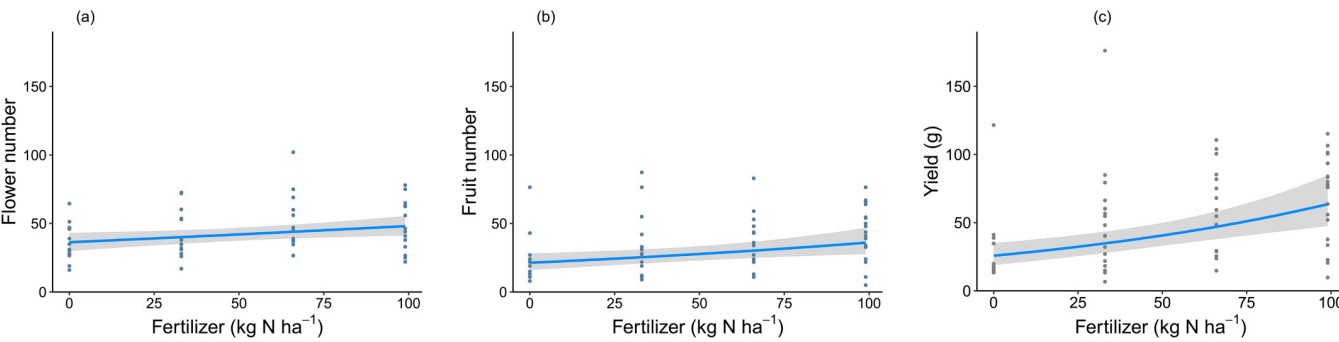

**Fig 1.** Effects of fertilizer application rates on flower number (a), fruit number (b) and yield (c) per plant. Graphs show conditional partial regression plots based on the minimum adequate models. Shadings show the 95% confidence interval, and points represent partial residuals.

99 kg N·ha$^{-1}$ increased yield from 25.7 g to 63.5 g (Fig 1C). SOM content or AMF inoculation did not affect these yield parameters, nor did they influence the effect of fertilizer (no significant interactions; Table 1).

A three-way interaction was found between the effects of AMF inoculation, SOM content and fertilizer inputs on fruit set (Table 1). In low SOM soils, the fruit set increased with increasing fertilizer inputs, for both AMF inoculated and non-inoculated plants (Fig 2A). In high SOM soils, the fruit set of non-inoculated plants showed a positive relationship with fertilizer inputs, while the fruit set of inoculated plants was highest in unfertilized soils and decreased with increasing fertilizer inputs (Fig 2B).

There was also a three-way interaction between the three experimental variables on the single berry weight per plant (Table 1). In low SOM soils, the relationship with fertilizer application rate was much more pronounced for AMF inoculated plants than for non-inoculated plants (Fig 3A). In high SOM soils, single berry weight was consistently higher in AMF inoculated plants than in non-inoculated plants, although the difference seemed to decrease with increasing fertilizer inputs (Fig 3B).

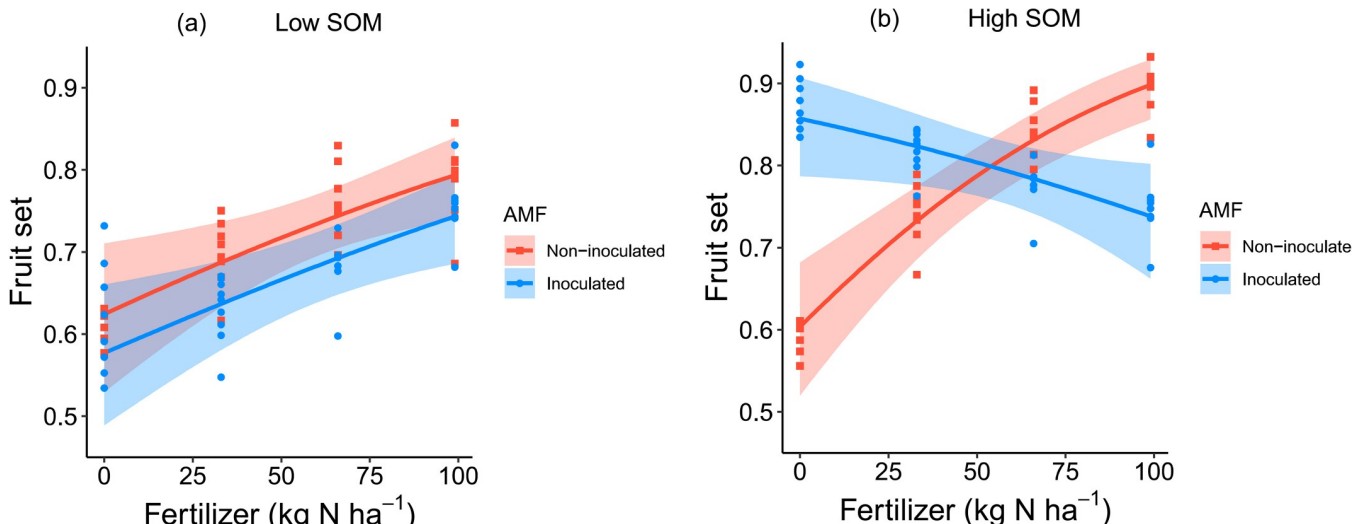

**Fig 2. Interactive effects of AMF inoculation, SOM and fertilizer application rates on fruit set per plant.** Graphs show conditional partial regression plots based on the minimum adequate model; shadings show the 95% confidence interval, and points represent partial residuals.

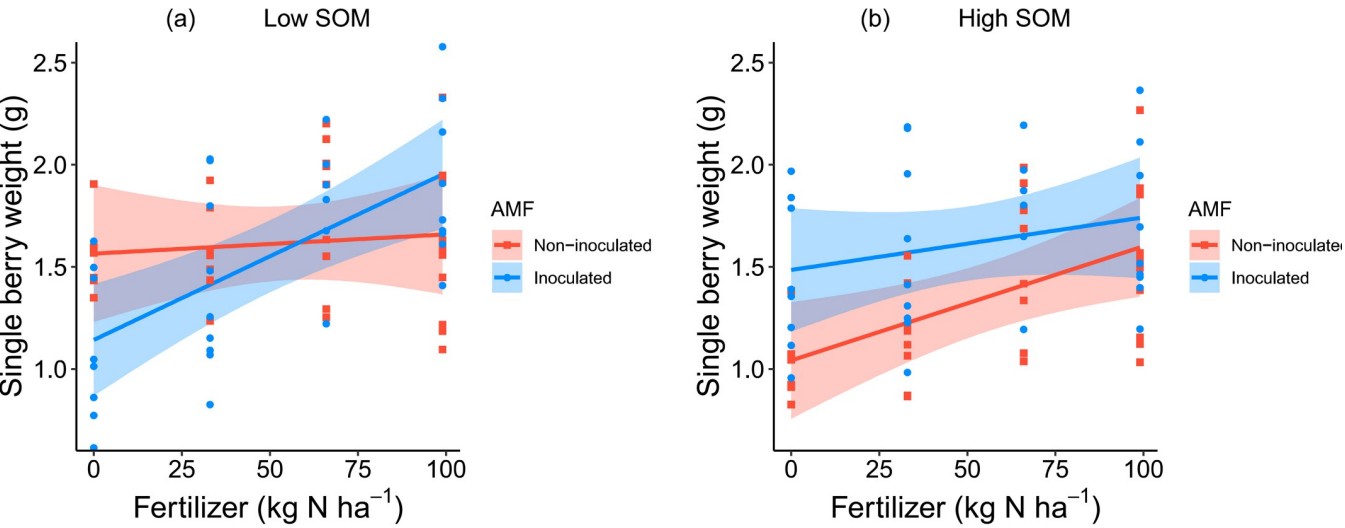

**Fig 3. Interactive effects of AMF inoculation, SOM and fertilizer application rates on average single berry weight (g) per plant.** Graphs show conditional partial regression plots based on the minimum adequate model; shadings show the 95% confidence interval, and points represent partial residuals.

## Discussion

In this study, we found that the numbers of flowers and fruit, as well as the most important parameter from the perspective of farmers, yield per plant, were only driven by fertilizer inputs and were not significantly impacted by AMF inoculation or SOM content. The positive relation between fertilizer application and fruit set in AMF inoculated plants in low SOM soils, changed into a negative relation in high SOM soils. Similarly, at low SOM the relation between fertilizer application and single berry weight was more pronounced in inoculated plants than in non-inoculated plants, but in high SOM soils it was the other way around. This suggests that the effects of AMF and SOM on these yield parameters cancel each other out and as a result did not contribute to the final yield.

At first glance, the lack of effects of AMF inoculation and SOM content on yield, flower number and fruit number, may seem at odds with results of earlier studies done using this same study system. For example, Chen, Kleijn (52) found significant positive effects of AMF inoculation on raspberry flower number, fruit number and yield. Furthermore, in wild raspberry populations, we found that yield was positively related to the SOM content (Chen K. et al., unpublished results, January 2022). However, the first study was only done at low SOM content levels (1.95%; the same as the current low SOM content treatment), while the second study was exclusively done in high SOM content soils (mean 7.4%, range 3.2–13.1%), and neither of these studies simultaneously manipulated both SOM content and AMF inoculation. Potential tradeoffs between the effects of the two factors on raspberry yield would therefore not become apparent in these studies. This is further supported by the fact that Fig 3A is almost an exact copy of Fig 3 in [52]. Both these graphs show the effects of fertilizer and AMF on single berry weight under the same low SOM content levels. Furthermore, in our previous experiments we showed that part of the effects of AMF and SOM could be explained by their positive influence on flower visitation rate by pollinators [52,57]. Because pollinators were not considered in this study, this may have left unexplained any potential indirect effects of AMF inoculation and SOM content on flower visitation rate and consequently the final yield.

In low SOM soils, AMF-inoculated plants produced smaller raspberries than the non-inoculated plants under low fertilizer inputs, while the beneficial effects of AMF inoculation on

berry weight only became apparent at adequate fertilizer inputs. One possible explanation for this is that under nutrient deficiency AMF have to compete for the limiting nitrogen for their hyphae development against the host plants, reducing the resources that host plants can allocate to fruit development [58,59]. In high SOM soils and under low fertilizer inputs, AMF inoculation increased both single berry weight and fruit set compared to those of the non-inoculated treatments (Figs 2B and 3B), likely because AMF could help acquire nutrients from soil organic matter to compensate for the effects of artificial inputs [60]. However, the benefits of AMF inoculation tended to decrease (Fig 3B) or even change into parasitic effects (Fig 2B) with increasing fertilizer application rate, a pattern found in previous studies as well [61–63]. The demonstrated tradeoff between effects of AMF inoculation and SOM content at different fertilizer application rates on berry weight and fruit set might explain why we didn't observe any effect of these factors on flower number, fruit number and yield. The negative interaction could be explained by the cost-benefit relation between AMF and the host plants [61–63]. Host plants share up to 20% of total photosynthetic carbon with AMF, as the cost to maintain the symbiotic associations [64], while receiving mineral nutrients and other resources absorbed by AMF as the benefit [29]. The cost-benefit relations vary from positive to negative, depending on the environmental context and the identity of AMF and the host plants [65]. Under high fertilizer inputs and high SOM soils, the host plants might obtain adequate nutrients via their own root systems [66,67], which decreases the dependence on the assistance of AMF over nutrients acquisition. However, if the associated cost does not decrease, or less strongly, this may result in a net negative benefit which may explain the decreasing benefits of AMF for single berry weight and fruit set with increasing fertilizer levels in the present study. In addition, the decreasing benefits of AMF inoculation might also be explained by the direct suppressing effects of the host plants on AMF growth. When plants obtain sufficient nutrients and water via their own root system in high nutrients soils (high SOM and fertilizer in this study), they may suppress AMF development [68]. Consequently, the suppressed AMF contributed less to production and this might indirectly constrain the benefits delivery of SOM since AMF can enhance the decomposition of SOM [69,70].

Although our study is based on only one study in one crop species, it is the first one to explore the interactive effects of AMF and SOM under a range of fertilizer application rates. Our results provide an indication that the benefits of AMF and SOM on crop yield offset each other. This finding contributes to the understanding of the dynamic effects of AMF inoculation on crop production. For example, Yamawaki, Matsumura [71] found significant positive effects of AMF inoculation on turmeric (*Curcuma longa* L.) production under greenhouse conditions but no effects were found under field conditions, and they attributed the differing outcomes to the influence of indigenous AMF. However, the lack of beneficial effects of AMF inoculation under field conditions could also be caused by tradeoffs due to the interactive effects between AMF, SOM and fertilizer, according to our findings. Therefore, our findings may have important implications for applying AMF as biofertilizers in practical cropping systems, which has been increasingly proposed as a key solution for accomplishing sustainable agriculture [72,73]. For example, when SOM content is high, inoculating AMF might not be such a good idea as when SOM content is low, unless with reduced fertilizer inputs. This study starts the exploration of the combined effects of AMF and SOM on raspberry production under several fertilizer inputs, and further research over a wider range of contexts (e.g. crop, soil type, climate, irrigation and fungicides) is needed to identify their interactive effects under real-world conditions.

## Supporting information

**S1 Table. The number of replicated raspberry plants per treatment combination.**
(PDF)

**S1 Data.**
(XLSX)

**S1 File. R code for the data analysis.**
(R)

## Acknowledgments

We thank Emiel van Riet for his assistance with the fieldwork.

## Author Contributions

**Conceptualization:** Jeroen Scheper, Thijs P. M. Fijen, David Kleijn.

**Formal analysis:** Ke Chen, Jeroen Scheper, Thijs P. M. Fijen, David Kleijn.

**Funding acquisition:** David Kleijn.

**Investigation:** Ke Chen, David Kleijn.

**Methodology:** Ke Chen, Jeroen Scheper, Thijs P. M. Fijen, David Kleijn.

**Project administration:** Ke Chen, David Kleijn.

**Supervision:** Jeroen Scheper, Thijs P. M. Fijen, David Kleijn.

**Validation:** Jeroen Scheper, Thijs P. M. Fijen, David Kleijn.

**Visualization:** Ke Chen, Jeroen Scheper, Thijs P. M. Fijen, David Kleijn.

**Writing – original draft:** Ke Chen.

**Writing – review & editing:** Ke Chen, Jeroen Scheper, Thijs P. M. Fijen, David Kleijn.

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
