## [Decision Letter · Decision Letter 0]

16 Mar 2022

PONE-D-22-02818Potential tradeoffs between effects of arbuscular mycorrhizal fungi inoculation, soil organic matter content and fertilizer application in raspberry productionPLOS ONE

Dear Dr. Chen,

Thank you for submitting your manuscript to PLOS ONE. After careful consideration, we feel that it has merit but does not fully meet PLOS ONE’s publication criteria as it currently stands. Therefore, we invite you to submit a revised version of the manuscript that addresses the points raised during the review process.

Four reviewers assessed the ms and raised important concerns, especially the statistical analysis, that I warmly suggest you to address.

We look forward to receiving your revised manuscript.

Kind regards,

Sergio Saia, Ph.D.

Academic Editor

PLOS ONE

Journal Requirements:

(We thank Emiel van Riet for his assistance with the fieldwork. KC was funded by the China Scholarship Council (File No. 201706990023).)

(The funders had no role in study design, data collection and analysis, decision to publish, or preparation of the manuscript.)

Reviewers' comments:

Reviewer's Responses to Questions

**Comments to the Author**

1. Is the manuscript technically sound, and do the data support the conclusions?

Reviewer #1: Yes

Reviewer #2: Yes

Reviewer #3: No

Reviewer #4: Partly

2. Has the statistical analysis been performed appropriately and rigorously? 

Reviewer #1: Yes

Reviewer #2: No

Reviewer #3: Yes

Reviewer #4: I Don't Know

3. Have the authors made all data underlying the findings in their manuscript fully available?

Reviewer #1: Yes

Reviewer #2: Yes

Reviewer #3: Yes

Reviewer #4: No

4. Is the manuscript presented in an intelligible fashion and written in standard English?

Reviewer #1: Yes

Reviewer #2: Yes

Reviewer #3: Yes

Reviewer #4: Yes

5. Review Comments to the Author

Reviewer #1: It is interesting to study the interactive effects of SOM and AMF on the production of raspberry, the results showed that potentially beneficial effects of AMF and SOM can be offset by each other, probably driven by the dynamic relations between AMF and the host Plants.

In materials and methods, pH value in soil, and other mineral nutrients such as Ca, Mg, Zn and Cu etc in soil needs to be supplemented. Additionally, the AMF-inoculated dosage, the spore number, and the mycorrhizal colonization determination method needs to be supplemented too.

Reviewer #2: The manuscript aim is to study the interaction between soil organic matter content (SOM) and arbuscular mycorrhiza fungi (AMF) inoculation and how they are affected by mineral fertilization application in raspberry yield. The hypothesis of the work is there is a tradeoff between SOM and AMF. The objectives are interesting and novel because the authors studied the three factors interaction: SOM, AMF, and mineral fertilizer.

The methodology is well written and honest because the authors explain that they have problems with plant survival and growth. However, the experimental design and data analysis have to be improved because the description of the blocks is not understandable. Looking at the excel data file of the supplementary material, the blocks hold different treatments and are not balanced. For example, blocks 1, 2, and 3 contain all the treatments (AMF, SOM, and fertilizer) for big plants. However, block 5 contain only inoculated, low SOM and 2 fertilizers for small and block 6 inoculated plants and low SOM but 1 plant of each fertilizer level. In my opinion, blocks have to contain all the treatments.

In line 172-173, the authors say that “response variables were averaged per plant to avoid pseudoreplication”. But it is only possible in the single berry weight. The authors calculated the sum of all the values of the other variables, yield, flower, and fruit number per plant.

The discussion part and the conclusions are well written.

The reference 58, the journal is Acta Horticulturae, and in reference 69 the journal name is missed.

In Figure 1, the graph only shows plant response to fertilizer level, however, in the caption effects of AMF and SOM are included. This caption need revision.

Reviewer #3: The major limitation of this study is that AMF colonization in the roots of Rubus idaeus was not assessed. The authors conclude about the effects of AMF, however, without the data on AMF colonization there is no guaranty that AMF inoculation actually resulted in the formation of arbuscular mycorrhizas. Without mycorrhizas there can be no effects. Mycorrhization depends on multiple factors, such as quality of inoculum, target plant species, environmental factors, etc., so it cannot be taken for granted. AMF colonization assessment needs to be done.

Reviewer #4: The manuscript reports data on a mixed experimental setup (plants of two different origins), without a detailed description of the experimental design:

- the authors claim that they used a randomized complete block design, but no explanation of the variables on which blocks were designed and why is provided;

- the data provided reveal that unbalanced block design is obtained, and many blocks lack most of the treatment. as my expertise in R is limited, I cannot check it by myself, thus I ask the authors if the linear mixed effects models take in account this uneven distribution of treatments

- it seems to me that one single plant per treatment per block (when available) has been monitored: what does it mean that "Response variables were averaged per plant to avoid pseudoreplication", given that data originate from only one plant? please clarify

- data provided in the excel file report only the variable "weight", assessed for each data point, although in the text other variables are analysed (flower number, fruit number, single berry weight (g/fruit) and total yield (g/plant)). As the journal requires authors to make all data underlying the findings described in their manuscript fully available without restriction, I think results fo all variables should be provided

The absence of data and details make this manuscript difficult to be properly evaluated.

6. PLOS authors have the option to publish the peer review history of their article (what does this mean?). If published, this will include your full peer review and any attached files.

Reviewer #1: No

Reviewer #2: No

Reviewer #3: No

Reviewer #4: No

---

## [Author Response · Author response to Decision Letter 0]

22 Apr 2022

Response to Reviewers

Reviewer #1: It is interesting to study the interactive effects of SOM and AMF on the production of raspberry, the results showed that potentially beneficial effects of AMF and SOM can be offset by each other, probably driven by the dynamic relations between AMF and the host Plants.

In materials and methods, pH value in soil, and other mineral nutrients such as Ca, Mg, Zn and Cu etc in soil needs to be supplemented. Additionally, the AMF-inoculated dosage, the spore number, and the mycorrhizal colonization determination method needs to be supplemented too.

Reply: Thanks for the reviewer's suggestions. We have added pH values of the experimental soils in the revised manuscript in line 112 and 114. Unfortunately, we don’t have measures of Ca, Mg or Zn. As we used the same soils for the experiment, and we focus on the effect of artificial fertiliser (N, P and K), we think those nutrients have limited effects on the final results. We added the spore number of the AMF inoculum in line 115, and the dosage was added in line 124. 

In this study we have focussed on the management practice of AMF inoculation, rather than studying the effect of AMF on yield. In an earlier study we found that all plants were colonised similarly by AMF at the end of the experiment, likely because the plants were growing already for almost a year in non-sterilised soils (Chen et al 2022, DOI: 10.1016/j.agee.2021.107742). We nevertheless found (there and here) effects of AMF inoculation on yield, suggesting that the management practice of inoculation had an effect. Because we do find effects of AMF inoculation, and because we had limited access to the laboratory due to COVID-19, we decided not to measure root colonization for this study, and can therefore not add this information. However, we have now elaborated on this choice in the methods. 

Reviewer #2: The manuscript aim is to study the interaction between soil organic matter content (SOM) and arbuscular mycorrhiza fungi (AMF) inoculation and how they are affected by mineral fertilization application in raspberry yield. The hypothesis of th e work is there is a tradeoff between SOM and AMF. The objectives are interesting and novel because the authors studied the three factors interaction: SOM, AMF, and mineral fertilizer.

The methodology is well written and honest because the authors explain that they have problems with plant survival and growth. However, the experimental design and data analysis have to be improved because the description of the blocks is not understandable. Looking at the excel data file of the supplementary material, the blocks hold different treatments and are not balanced. For example, blocks 1, 2, and 3 contain all the treatments (AMF, SOM, and fertilizer) for big plants. However, block 5 contain only inoculated, low SOM and 2 fertilizers for small and block 6 inoculated plants and low SOM but 1 plant of each fertilizer level. In my opinion, blocks have to contain all the treatments.

Reply: Thanks for the reviewer's comments. We agree with the reviewer that it is unfortunate that we have incomplete treatment combinations due to relatively high mortality. The blocks were physical allocations of the pots to account for random gradients at the experimental site, and can therefore not be altered anymore. However, our used statistical methods (general linear mixed models) can cope with this unbalanced data quite well, by only looking at the available pairwise comparisons within block, and then generalising this pattern over all the blocks. But to make the unbalanced data more explicit, we added a supplementary table (line 159) in the revised manuscript, to indicate the exact number of replicated raspberry plants per treatment combination.

In line 172-173, the authors say that "response variables were averaged per plant to avoid pseudoreplication". But it is only possible in the single berry weight. The authors calculated the sum of all the values of the other variables, yield, flower, and fruit number per plant.

Reply: Thanks for the suggestion. We have clarified in lines 166-167 that only the single berry weight per plant was averaged. 

The discussion part and the conclusions are well written.

The reference 58, the journal is Acta Horticulturae, and in reference 69 the journal name is missed.

Reply: Thanks. We have added the journal names in both references. In the current version, the original reference 58 is now reference 51 (reference style is changed to meet requirements of the journal); and reference 69 is now reference 71.

In Figure 1, the graph only shows plant response to fertilizer level, however, in the caption effects of AMF and SOM are included. This caption need revision.

Reply: Thanks for the comments. We have revised the caption.

Reviewer #3: The major limitation of this study is that AMF colonization in the roots of Rubus idaeus was not assessed. The authors conclude about the effects of AMF, however, without the data on AMF colonization there is no guaranty that AMF inoculation actually resulted in the formation of arbuscular mycorrhizas. Without mycorrhizas there can be no effects. Mycorrhization depends on multiple factors, such as quality of inoculum, target plant species, environmental factors, etc., so it cannot be taken for granted. AMF colonization assessment needs to be done.

Reply: Thanks for the reviewer's comments. As also explained in a response to reviewer 1: In this study we have focussed on the management practice of AMF inoculation, rather than studying the effect of AMF on yield. Indeed, in an earlier study we found that all plants were colonised similarly by AMF at the end of the experiment, likely because the plants were growing already for almost a year in non-sterilised soils (Chen et al 2022, DOI: 10.1016/j.agee.2021.107742). We nevertheless found (there and here) effects of AMF inoculation on yield, suggesting that the inoculation had an effect. Because we do find effects of the management practice of AMF inoculation, and because we had limited access to the laboratory due to COVID-19, we decided not to measure root colonization for this study, and can therefore not add this information. However, we have now elaborated on this choice in the methods. 

Reviewer #4: The manuscript reports data on a mixed experimental setup (plants of two different origins), without a detailed description of the experimental design:

- the authors claim that they used a randomized complete block design, but no explanation of the variables on which blocks were designed and why is provided;

Reply: Thanks for the reviewer's comments. Each block in the randomized complete block design (RCBD) of the experiment was designed to consist of all treatment combinations. However, only a few of the plants survived the fruit production process or survived but did not produce fruits, and thus the final data were unbalanced. In the revised manuscript, we have clarified this, and we added a supplementary table (line 159) to indicate the number of replicated raspberry plants per treatment combination.

- the data provided reveal that unbalanced block design is obtained, and many blocks lack most of the treatment. as my expertise in R is limited, I cannot check it by myself, thus I ask the authors if the linear mixed effects models take in account this uneven distribution of treatments

Reply: Thanks for raising this. Yes, our statistical approach can deal well with these unbalanced data, because it first compares effects within block and then generalises effects over blocks (Zuur et al. 2009, ISBN:9780387874586). These methods have also been used in similar experiments with unbalanced data (e.g., Motzke et al. 2015, DOI: 10.1111/1365-2664.12357; Tamburini et al. 2015, DOI: 10.1007/s00442-015-3493-1 )

- it seems to me that one single plant per treatment per block (when available) has been monitored: what does it mean that "Response variables were averaged per plant to avoid pseudoreplication", given that data originate from only one plant? please clarify

Reply: Thanks for spotting this mistake. We have clarified in lines 166-167 that it is the single berry weight per plant that was averaged. 

- data provided in the excel file report only the variable "weight", assessed for each data point, although in the text other variables are analyzed (flower number, fruit number, single berry weight (g/fruit) and total yield (g/plant)). As the journal requires authors to make all data underlying the findings described in their manuscript fully available without restriction, I think results fo all variables should be provided

The absence of data and details make this manuscript difficult to be properly evaluated.

Reply: Thanks for noticing this omision. We have added data of all variables in the revised excel file.

---

## [Editor Report · Decision Letter 1]

27 May 2022

Potential tradeoffs between effects of arbuscular mycorrhizal fungi inoculation, soil organic matter content and fertilizer application in raspberry production

PONE-D-22-02818R1

Dear Dr. Chen,

We’re pleased to inform you that your manuscript has been judged scientifically suitable for publication and will be formally accepted for publication once it meets all outstanding technical requirements.

Kind regards,

Sergio Saia, Ph.D.

Academic Editor

PLOS ONE
---

## [Editor Report · Acceptance letter]

8 Jul 2022

PONE-D-22-02818R1 

Potential tradeoffs between effects of arbuscular mycorrhizal fungi inoculation, soil organic matter content and fertilizer application in raspberry production 

Dear Dr. Chen:

I'm pleased to inform you that your manuscript has been deemed suitable for publication in PLOS ONE. Congratulations! Your manuscript is now with our production department. 

Kind regards, 

on behalf of

prof Sergio Saia 

Academic Editor

PLOS ONE